# Towards Cultivating Decentralised Data Privacy, Interoperability and Trust with Semantic PETs and Visualisations[*]

Anelia Kurteva[1,*,†], John Domingue[2,†]

[1]*Delft University of Technology, The Netherlands*
[2]*Knowledge Media Institute (KMi), The Open University, The UK*

## Abstract

Recent artificial intelligence (AI) advancements in the fields of generative AI and hyper-automation in the Internet of Things (IoT) have turned data into a valuable highly sought-after asset and an economical resource for the ever-growing service digitization. Fields such as smart cities, e-commerce and finance now often integrate AI to improve and optimise online services, which requires large volumes of diverse high-quality data. Most of the data that is generated, related and used by humans for AI in any of these domains can be categorised as personal. The access to it, its processing and sharing for different purposes between different software agents, humans and organisations, if not governed and legally compliant, can jeopardise individuals' privacy and sovereignty both online and offline. Through the years, several eminent data misuse cases have shown that the current centralised digital data ecosystem is easily exploitable and that there is a lack of transparency and accountability along the data supply chain. Individuals have long ago lost control over their data due to vendor lock-ins and their privacy is often violated. The growing number of fines issued to numerous organisations in response to violating the General Data Protection Regulation (GDPR) by misusing individual's personal data, further confirm this. A new paradigm shift towards decentralisation of the Web has emerged as a solution. However, implementing data and privacy governance in a decentralised setting poses new technical and organisational challenges that are currently being investigated and a standard solution is yet to be established. Further, there is a lack of tools aimed at assisting and guiding individuals in managing their decentralised data. In this paper, we propose the development of a more human-centered approach for building trusted self-sovereign decentralised spaces for personal data governance based on combining semantics with privacy enhancing technologies (PETs) and the utilisation of graphical visualisations. We present the main building blocks of the proposed approach with the main goal to foster further discussion and collaboration between the Semantic Web, Privacy, Decentralisation, Human-Computer Interaction and Legal communities.

## Keywords

Decentralisation, Privacy Enhancing Technologies, Ontologies, Knowledge Graphs, Data Visualisation, Legal Compliance, Trust

## 1. Introduction

The data economy is expected to rise to staggering 827 billion euros in value by 2025 [1]. According to the European Commission, *"Data is the lifeblood of the economy and a driver of innovation"*[1]. However, the access, processing and sharing of the data for different purposes between different software agents, humans and organisations if not governed and legally compliant, can jeopardise individuals' privacy and sovereignty both online and offline. Prominent examples of data misuse cases are National Security Agency's mass intelligence-gathering surveillance programs [2], Cambridge Analytica's data harvest [3] and Clearview's[2] Artificial Intelligence (AI) social media image collection[3]. Over the past few years, the risks to one's privacy and personal data sovereignty, stemming from vendor lock-ins due to the current quasi-monopolistic data economy [4], have motivated a new paradigm shift towards the

---

*NXDG: NeXt-Generation Data Governance, SEMANTiCs 2024, Amsterdam, The Netherlands*

[*]Corresponding author.
[†]These authors contributed equally.
✉ a.kurteva@tudelft.nl (A. Kurteva)

[1]https://digital-strategy.ec.europa.eu/en/library/building-data-economy-brochure
[2]https://www.clearview.ai
[3]https://www.theguardian.com/technology/2022/may/23/uk-data-watchdog-fines-facial-recognition-firm-clearview-ai-image-collection

decentralisation of data on the Web. Endorsed by the creator of the Web himself, decentralisation aims to make individuals *"once again be the masters of their own data"*[4] by separating data from services [5]. Further discussion on decentralisation is provided in [6]. The need for individuals' data empowerment has also led to the enforcement of laws such as the EU's GDPR [7]. However, the research thus far has focused primarily on advancing the technology itself and has overlooked the human-centered side. Non-experts, whom decentralisation aims to empower, are in need of easy to understand and use, informative user interfaces (UIs) that simplify and minimise the burden of decentralised data governance [8][9]. Currently, only one such UI [9] has been designed and published as an open-source. Guidelines for implementing GDPR-compliant decentralised data governance, which individuals can follow, are yet to be defined as well [8][10]. Having investigated existing related work on privacy, semantics, human comprehension and data governance in (but not limited to) decentralised settings (e.g [10][8][9][11][12][13]) we have identified the following five challenges that we believe limit its further adoption. Challenge 1 (**C1**)-supporting decentralised data interoperability between different agents (humans, machines), stems from the complexity of data and process management across decentralised agents and the need for unified vocabulary to catalogue data sharing which is used as a standard (further elaborated in [10]). This introduces challenge 2 (**C2**) - establishing responsibility and fostering accountability across decentralised agents. An agent's identity and role(s) should be clearly defined and verified as each agent can have different roles in different use cases thus various responsibilities. The lack of data interoperability and unclear responsibilities affect the transparency of a system and end-users' trust in it. Legally compliant data sharing and processing based on one's informed consent is a necessity and a building block of trust. However, it is still not clear how to best support individuals in making sense of decentralised data sharing and the consent for it (viewed as **C3**). This topic is further discussed by the authors in [10][8][9]. The above mentioned challenges further relate to privacy-preservation (another building block of trust). How can we support decentralised data interoperability and process transparency (e.g. clearly establish responsibilities, verify agents' identities) while preserving privacy? Following this, we define **C4** - ensuing sensitive decentralised personal data is protected and only shared with verified agents in a privacy-preserving manner. Last but not least, based on discussions on the performance of decentralised web technology (e.g. [14][15][16]) we define performance as challenge **C5**. To address these challenges and to help cultivate a trusted data economy, we propose the DataPrInTs approach - an interdisciplinary human-centered approach to decentralised data governance based on the combination of Privacy Enhancing Technologies (PETs)(i.e. tools or technologies aimed at enhancing privacy [17]) and semantics (i.e. ontologies and knowledge graphs) and added data visualisations such as user interface(s) (UI) for decentralised data flows and consent management. In this context, we view trust as a *"firm belief in the reliability, truth, or ability of someone or something"*[5][13]. The main goals of this approach are to:

- Establish trust in data spaces through visualisations that raise transparency of decentralised data sharing flows between data spaces and all actors in them
- Assist individuals in making sense of decentralised data sharing by using visualisations as a tool for privacy explanations
- Explore incentives for decentralised data sharing
- Define machine-readable decentralised data sharing policies, licenses and contracts that support legal compliance
- Foster data exchange between data spaces while preserving individuals' privacy

Section 2 outlines the proposed approach and its building blocks. A proposal for next steps towards the approach's implementation for two use cases (i.e. AI for sustainability and education) are presented in Section 3. Conclusions can found in Section 4.

---

[4]https://www.inrupt.com/blog/flanders-solid
[5]https://www.merriam-webster.com/dictionary/trust

## 2. Towards an Approach for Decentralised Personal Data Privacy, Interoperability and Trust

Currently in centralised systems service providers are responsible for storing and processing individuals' data in a legally compliant way. In a decentralised system, individuals are given control and ownership of their data, which can be a burden [10][8]. While in favour of sovereignty, this promotes an unrealistic expectation that individuals are well aware of the applicable laws, their rights and have a level of privacy knowledge that can help them make informed decisions about their personal data management in decentralised settings. Building a trusted and privacy-preserving decentralised ecosystem requires an interdisciplinary approach that combines knowledge from the Semantic Web, Legal, Privacy, Human-computer Interaction and even AI domains. The following sections present our proposal for such approach and its main building blocks.

### 2.1. DataPrInTs Approach Proposition

Following the presented in [10] and in previous sections challenges to decentralised data governance, privacy and trust, we propose the following interdisciplinary human-centered approach (see Fig. 1) for preserving decentralised personal data privacy and establishing data interoperability and individuals' trust.

Semantic Web technologies, including ontologies and knowledge graph can be used to support the majority of findable, accessible, interoperable, reusable (FAIR) [18] data principles (e.g. see box 2 in [18] for principles *"F2. Data are described with rich metadata"*, *"A1. (Meta)data are retrievable by their identifier using a standardised communications protocol"*). Ontologies can define a semantically rich schema of personal data spaces, the data they safeguard and access and usage policies related to them. Regarding privacy preservation, the combination of PETS such as differential privacy [19], multi-party computation [20], federated learning [21] with semantic-based data access and usage mechanisms can facilitate a more-context aware data processing and privacy-preservation. This can be extremely useful for making AI privacy-aware in sensitive cases such as personal medical treatment or health insurance policy recommendation.

Data visualisations are a key tool to raise individuals' awareness and ease their comprehension of decentralised data flows thus help cultivate more trust. Visualisations (e.g. UIs) can also be used as a communication channel for privacy explanations to individuals. Taking a step further, to better understand individual's motivation to participate in a decentralised data ecosystem, smart contracts and licences for data sharing that incorporate incentives can be explored as well. Following the proposed approach, a proof-of-concept prototype based on privacy-by design principles (e.g. user-centric, data minimisation, system and process transparency) in the form of a software tool with an interface that guides individuals through their personal decentralised data ecosystem will be implemented. The tool's evaluation (both qualitative and quantitative) in terms of its usability, ability to ease individual's comprehension of decentralisation and increase trust in data spaces will help gather valuable insights on individuals' perspectives of decentralisation and privacy. Further, the tool can be evaluated in terms of its ability to support decentralised data audits and legal compliance.

### 2.2. Approach Building Blocks

The following sections present in more detail the main legal, technical and human-focused building blocks of the proposed approach. In addition, for each building block, we propose a concrete utilisation and propose several (research) questions that come into light for further discussion on the topic(s).

#### 2.2.1. Legislation

Several legislations aimed at fostering stronger data protection that empowers individuals, for example, the GDPR [7], the Data Act [22], the Data Markets Act [23] and the European Data Governance Act

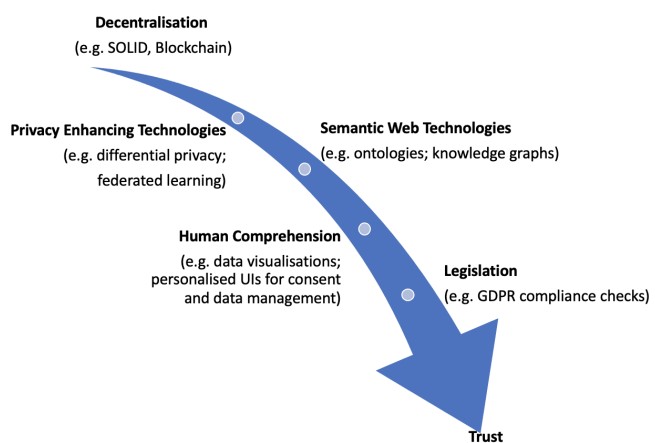

**Figure 1:** DataPrInTs's Trust Building Blocks. Decentralisation is viewed as key approach to facilitating privacy-preserving and trusted data sharing. PETs that use semantically enriched data and are context aware are utilised to support data privacy and security. Semantic Web technologies support data interoperability. Easing individuals' comprehension of decentralisation and personal privacy is achieved with the help of data visualisations (e.g. UIs) as key communication channel. Compliance verification of how the system was built and of its currently running processes with legislation such as the GDPR, the AI Act are essential to build up trust as well.

[24] have already been or are yet to be enforced. Preserving an individual's privacy has become a key objective for ensuring legal compliance. However, with the current developments now rising to the fore in the digital economy, for example, service digitization in the IoT sector and AI advancements, a discrepancy between the law and its technological application can be observed. An example of this is the application of GDPR's principles for data protection to decentralisation, which has a different approach to data sharing as it puts the responsibility in the hands of the individuals [10]. More recent legislations such as the AI Act [25] needs to be considered as well when developing AI and utilising PETs such as federated learning (i.e machine learning over remote data sources) [26].

Having this in mind, we believe that the problem needs to be investigated through both technology and legal perspectives. A set of requirements (aligned with the law(s)) that can be used as guidelines for GDPR-compliant trusted personal data governance in decentralised settings can be derived and documented in a machine-interoperable format to ease automation (e.g. of compliance and auditing) when needed. The added benefits of semantics for automated GDPR compliance verification have already been showcased by the authors in their previous work in [27][28]. Last but not least, clashes and overlaps between legislations regarding data protection and its use for AI need further investigation which might need to be use case specific.

### 2.2.2. Decentralisation

The shift to decentralisation has slowly but successfully started to take place[6,7]. Decentralised identifiers (DIDs)[8], Distributed Ledger Technology (DLT)[9] and personal data stores such as Solid [29], have emerged as decentralised technologies that enhance privacy, enable security and process transparency and help individuals regain ownership of their data. However, decentralisation has also shifted the roles, responsibilities and obligations of actors involved in data sharing. This affects how GDPR's legal basis, namely informed consent (Art. 7) and individual's rights (*"right to be forgotten"* (Art. 17(2))) are communicated to individuals and are enforced. The technology developments have focused mainly on

---

[6]https://www.cnbc.com/2020/11/09/tim-berners-lee-attracts-nhs-bbc-natwest-to-inrupts-solid-platform.html
[7]https://www.inrupt.com/blog/flanders-solid
[8]https://www.w3.org/TR/did-core/
[9]https://assets.publishing.service.gov.uk/government/uploads/system/uploads/attachment_data/file/492972/
gs-16-1-distributed-ledger-technology.pdf

the back-end leaving behind the front-end interfaces that are the mediums end-users need to interact with their decentralised data. *"The development of an intuitive user experience is of the highest importance to SolidLab"*[10]. Recent UI research [9] has shown promising results, has further highlighted the need for visual explanations of decentralised data sharing and the consent associated with it and has uncovered a new set of challenges related to individual's comprehension of decentralisation. One of our main goals is to define requirements (both functional and non-functional) that help design and implement an easily comprehensible UI that utilises dynamic data visualisations to assist individuals in making sense of their decentralised data sharing. These visualisations can also vary depending on the selected decentralised technology, context and individuals. However, an important thing to further investigate and discuss concerning this approach is the sustainability of the visualisations themselves.

### 2.2.3. Privacy Enhancing Technologies (PETs)

PETs not only enhance one's privacy by restricting and minimising data collection, the access to it and its usage and availability, but also by providing a level of security during data sharing through encryption [30]. According to Information Commissioner's Office (ICO)[11], PETs play a vital role in establishing effective data governance [31] and recommend their wider adoption in industry [32]. Most PETs operate on the principle of data minimisation, which while supporting privacy can limit the accuracy and explainability levels of automated AI-based decision making. There is a conflict between maintaining data privacy and its accessibility and interoperability that needs to be resolved to help achieve the future goals of the data economy [22][23][24]. The application of PETs in decentralised systems is also gaining traction. Several articles such as [33] explore the combination of PETs and blockchain [34]. However, due to its immutable nature, blockchain poses a risk to individuals' privacy and restricts GDPR's right to be forgotten [35][36][37]. The maturity level of different PETs (e.g. federated learning, multi-party computation, differential privacy etc.) and their suitability for utilisation in different decentralised data sharing contexts needs to be investigated. Guidelines in a machine-interoperable format can be provided and the process of a suitable PET recommendation can be automated with machine learning (trained on past PET success and failures and considering use case context). Within our approach, we plan to utilise ontologies and knowledge graphs to develop more context-aware PETs that support both data interoperability and GDPR's principles of transparency, traceability and data protection by design (Art. (25)). A challenge here is to balance data's privacy and security and its FAIRness.

### 2.2.4. Semantic Web Technologies

Ontologies and knowledge graphs stand out as two widely utilised semantic web technologies that support data interoperability, traceability and transparency [38]. Since the acceptance of the GDPR, these technologies have become the *"go-to"* solution for building structured, standardised, human- and machine-readable representations of and reasoning over legal knowledge [39][40]. The Data Privacy Vocabulary (DPV)[12], GConsent [41], Data Use Ontology (DUO) [42] and smashHitCore [43] are just some of the examples of ontologies focused on representing legal knowledge and supporting both machines and humans in making sense of it. More recent work on the topic has been carried out in the scope of the smashHit[13] project, which has developed knowledge graph-based mechanisms for automated consent [27] and contract [28] compliance verification for smart city and insurance-focused sensor data sharing. The results have confirmed the benefits of semantics for process explainability and optimised decision making. However, all these studies have focused on the challenge of performing GDPR-compliance verification in centralised settings. In our case, we plan to explore how semantics can be used to enforce GDPR in decentralised data sharing contexts and to provide clear specifications of each decentralised actor's roles in order to establish responsibility, ensure accountability and build trust.

---

[10]https://solidlab.be
[11]https://ico.org.uk
[12]https://w3c.github.io/dpv/dpv/
[13]https://smashhit.eu

### 2.2.5. Human Comprehension of Data Sharing

Requesting and revoking informed consent in a GDPR-compliant manner, has turned out to be a significant challenge for many organisations [27]. An undeniable challenge to this are also the individual's comprehension needs and awareness of the possible implications of blindly giving consent [44]. Research [45][46] has shown that individuals are often unaware of what giving consent means and the implications that follow. Helping individuals make sense of data sharing and the consent for it through visualisations has been the focus of several studies (some of which this researcher has been part of), namely [47][48][49][50][51]. However, there has been limited work on how, when (prior to or post-consent has been granted) and what types of visualisations can be utilised to most effectively aid individuals' comprehension of decentralised data sharing [9]. Our prior research on data visualisations to aid consent [50][49] and web cookies [52] comprehension has shown that different individuals have different comprehension needs when it comes to their data sharing and legal rights thus we plan to investigate various tools (including Generative AI such as DALLE[14], Midjourney[15]) for personalised and dynamic on-the-go data visualisation generations. An important perquisite is to know who the end-user is, what type of comprehension needs they have and the context of data sharing.

## 3. Use Case Exploration and Next Steps

We have set to investigate several use cases for the implementation of this approach. Specifically, we have identified two suitable use cases. Both of these use cases demonstrate the complexity and interplay between legislation and technology (need for FAIR data and privacy-preservation).

Use case 1 (UC1) focuses on data sharing for building digital product passports (DPPs) of personal ICT devices (e.g. laptops, tablets, smartphones) that are major stream of focus in the Circular Economy (CE) due to their increasing impact on the environment in terms of e-waste and need for critical materials [53] [54]. A DPP can be viewed as collection of data about a device captured through its lifetime (from material mining for manufacturing, use, end-of-life etc.) stored in structured and machine-interoperable format [55]. In this use case, ICT data such as performance of the device at specific date, time and location, however, and can be classified as personal, which leads to privacy concerns. DPPs should be FAIR [18] but that should not be at the cost of privacy. Work on this has already begun on the Circular Resource Planning for IT Project (RePlanIT)[16], where the authors have build the RePlanIT ontology [56] for ICT DPPs. Decentralisation of the DPPs (e.g. Onto-DESIDE[17] project) is a possible solution that improves ones autonomy especially when the entity using the device is not the sole owner of it. For instance, DPPs for company-owned devices that are assigned to employees. The main challenges are to preserve privacy and support individuals' comprehension of decentralised data sharing for DPPs thus preventing mistrust due to a lack of process transparency and explainability.

Use case 2 (UC2) focuses on adopting decentralisation to facilitate trusted privacy-enhancing personal data sharing for cases such as personalised AI tutor systems in university settings. In 2020, 54% of UK universities reported a data breach [57]. Centralisation of the data has highlighted risks to students' privacy, who are often unaware of how and where their data is stored and managed, who has access to it and how it is used. Further, students lack control over the data itself. Decentralisation of personal data can help preserve privacy and support the transparency of current data sharing and processing practices within universities. The main challenges of this use case are raising awareness of decentralisation's benefits for students and the technical and human-centered implementation of decentralisation in a way that minimises the student's feeling of burden with regards to data governance. As next steps in the implementation of the proposed DataPrInTs approach, we have set up the following goals:

- Derive a set of requirements for trusted decentralised data sharing based on interviews, co-creation

---

[14]https://openai.com/index/dall-e-2/
[15]https://www.midjourney.com/
[16]https://www.ams-institute.org/urban-challenges/circularity-urban-regions/circular-resource-planning-for-it-replanit/
[17]https://ontodeside.eu

sessions and analysis of relevant research in the legal, technology and human behavioural domains; derive functional and non-functional requirements for data visualisations (e.g. UIs)

- Perform risk assessments for each use case; derive a set of technical and organisational measures; investigate suitable PETs, select semantics for each use case
- Identify existing relevant ontologies and reuse when possible to semantically model different context
- Ontologies, in combination with logic, can be used to represent decentralised data sharing policies and agreements in a standardised machine-interoperable format
- Design and implement interactive visualisations such as graphs, tables and forms; based on findings from co-creation sessions

## 4. Conclusions

In order to cultivate a trusted data economy in the future, digital infrastructures that promote data sovereignty, enable greater data protection and reinforce individuals' privacy awareness need to be implemented. Motivated by this, we presented an outline of the DataPrInTs approach for decentralised personal data privacy, interoperability and trust. Our interdisciplinary human-centered approach is grounded in the utilisation of semantics to make data interoperable, make PETs context aware, and of visualisations of consent and data sharing to ease individual's comprehension and raise awareness of personal data privacy in decentralised settings. Most importantly, with this paper, we aimed to highlight important factors such as privacy and legal compliance affecting one's trust in decentralisation and foster further discussion and collaboration between the Semantic Web, Privacy, Decentralisation, Human-Computer Interaction and Legal communities.

## 5. Acknowledgments

Anelia Kurteva is financially supported by the RePlanIT project funded by a Topsector Energy subsidy from the Ministry of Economic Affairs and Climate Policy in the Netherlands. The author would like to thank Ruud Balkenende and Alessandro Bozzon from TU Delft for their support and supervision.

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
