# OpenReview forum: "Towards Cultivating Decentralised Data Privacy, Interoperability and Trust with Semantic PETs and Visualisations"
_SEMANTiCS.cc/2024/Workshop/NXDG — NXDG 2024 Conditionallyed_

### Official Review · ~Rob_Brennan1 · 2024-07-30
**Intersting ideas, important topic, will stimulate debate but currently immature**

**Rating:** 6
**Confidence:** 4

**Review:**

Quality
There are many interesting arguments made about the future of decentralised data governance and the need for UIs and trust for end users. The state of the art is well referenced but perhaps a little too much weight is placed on grey literature. The work is ambitious.

There are a lack of definitions for many of the concepts discussed. The work is currently immature and contains many ideas for further work rather than solutions or even concrete plans.

Clarity
The paper is well written with a lack of typos or grammatical errors.
The lack of definitions means that it is not exactly clear what is being argued. Figure 1 seems to add little.

Originality and significance
There is a lack of work being carried out in this area.
The work is not currently very significant as it is early but it shows promise.
 In some ways the work is perhaps too ambitious so the scope is huge and this will make progress hard.

Detailed Comments

Abstract
"Fields such as smart cities, e-commerce, finance and medicine now often integrating AI to improve and optimise online services"
I suggest that you remove medicine from that list as there are very few actual deployments to date despite the hype.

"A new paradigm shift towards decentralisation of the Web has emerged as a solution."
How much of that is driven by blockchain marketing?
The Web is already a decentralised system. That is fundamental in the design of RDF. Please clarify with definitions of what you mean by centralised vs decentralised systems. I realise that this is often just code for "solid" vs "non-solid" but I think it would help your argument if you were more specific.

"However, the process of implementing data and privacy governance in a decentralised setting is yet to be defined."
GDPR already defines a decentralised data privacy governance regime at the organisational level. There is decentraised authority and decision-making. I think it would be good to refine this statement to clarify your intent.

"The main challenges limiting the
further adoption of decentralisation can be summarised as"
What about the performance implications of fine-grained decentralized architectures?
It would be good to have references for each of these challenges. For example I am not sure why privacy preservation is a specific challenge limiting adoption of  decentralisation.

"human-centered approach" - it is not clear what aspects of methods like socio-technical systems analysis or human factors ergonomics approaches to design you have adopted by having a focus on UIs/visualisation?
The focus on legal compliance is not really "human centred" either (despite being useful).
I think you need to come up with different terminology to describe your approach.

Typo: "liscences"-> "licenses"

Section 2

"In a decentralised system, individuals are given control and ownership of their data,"
I would say in some decentralised systems. But perhaps this is the definition of the type of system you mean when you say decentralised?

"Building a trusted and privacy-preserving decentralised ecosystem requires
an interdisciplinary approach that combines knowledge from the SemanticWeb, Legal, Privacy, Humancomputer
Interaction and even AI domains"

I suggest you need psychologists or social scientists too if you want trust.
It would be good if you defined what you mean by trust as there are many possible definitions.

"To ensure findable, accessible, interoperable, reusable (FAIR) [9] data principles are implemented,
SemanticWeb technology, ... can be utilised"

Just because you use ontologies/RDF, it does not guarantee FAIRness. I think you need to work harder than that and specify more. Eg check out a typical ontology in a FAIRness checker tool, it will probably not score well (despite having the capability to do so).

"Regarding privacy
preservation, the combination of PETS such as differential privacy, multi-party computation, federated
learning with semantic-based data access and usage mechanisms can facilitate a more-context aware
data processing and privacy-preservation."

Yes, they "can" facilitate it, but how? Also, do they create more risks? eg see how Federated Learning can leak private data https://ieeexplore.ieee.org/abstract/document/8737416
Your list is not very convincing, because it is just a list of possibilities. I think you need to have profiles or something to specify how.

"Following the proposed approach, a proof-of-concept prototype based on privacy-by design principles (e.g. user-centric, data minimisation, system and process transparency) in the form of a software tool with an interface that guides individuals through their personal decentralised data ecosystem will be implemented."
- There is a lot of things going on in there. Perhaps some prioritisation will be necessary?

"The tool’s evaluation in terms of its usability, ability to ease individual’s comprehension of decentralisation
and increase trust in data spaces will help gather valuable insights on individuals’ perspectives of
decentralisation and privacy."
How will you measure trust? (This is hard AFAIK)
"ease individual’s comprehension of decentralisation" sounds a bit suspect to me too. Normally users have internal models that are very different to the technological reality, but which are sufficient for them to make decisions - how will this influence your desire to make users comprehend decentralisation? (Should they seek to comprehend it exactly? Is that useful? NB this comes back to your definition of decentralisation too as it will be very useful for this.)

I cannot believe that you are citing a politically backed report by the Tory government into blockchain (DLT) as a justification for its success. We are now in 2024. Read this
https://www.sciencedirect.com/science/article/pii/S0160791X20303067

I wish you every success with this important work.

---

### Official Review · ~Kimberly_Garcia1 · 2024-08-01
**This paper presents DataPrInTs an approach that calls for interdisciplinary research to achieve data privacy in decentralized data stores.**

**Rating:** 3
**Confidence:** 5

**Review:**

This paper presents DataPrInTs, an approach that calls for interdisciplinary research in semantic technologies, data visualization, and PETs to create robust tools that build users trust in using decentralized data stores. The motivation of this contribution is strong, since it is true that decentralized data stores such as Solid are in their infancy, which means that the community is first looking into building infrastructure and ensuring the technical viability of the technology, rather than its practical day-to-day usage in real world scenarios. Even though, a few real-world use cases have been demonstrated, such as the Flanders government usage of Solid for managing diplomas and other academic documents, and a first demonstrator from the BBC; this is work in progress. Hence, a vast amount of research needs to be done before decentralized data stores can become an everyday commodity. Thus, research in making these technologies accessible, understandable, and easy to use, are of utmost importance. Even though the motivation highlights an important issue, the paper gets really hard to follow after its. Section 2 starts by introducing DataPrInTs without any background reference to the fields that this approach proposes to study in a collaborative manner. For example, in Page 2, there is a shallow statement about FAIR data, immediately followed by PETs, which are presented as a list of technologies, without references, or explanation on why we should care about differential privacy, multi-party computation, federated learning, etc. Indeed, Section 2.2.3 presents more detail on these technologies, but this comes too late, since in 2.1 the reader is left confused on why this list of technologies. The same happens with the second paragraph in page 3, which goes from data visualization (which is simplified to user interfaces) to smart contracts from one line to the other, without a clear narrative, or any detail on how they are connected in DataPrInTs.

Since the paper lacks a clear narrative on what is DataPrInTs and how this can be achieved, I would propose to restructure the paper by first presenting the building blocks, so the reader understand why these fields have been choosing, and then elaborating in depth Figure 1 (which is the main contribution of this paper), not only superficially with a list of technologies, but emphasizing what can be gained from each field. The narrative of this contribution could be made greatly strengthen if a use case is chosen and one aspect of each major field (semantics, data visualization, and PETs) is explained in depth to highlight how bringing these field together will contribute to the final goal of creating decentralized stores that are trusted by the users, since their data privacy will be preserved.

Other major issues:

Where do the four challenges presented in the introduction come from? Have they been found by the authors? If so, what was the analysis that allowed them to come to them, or is there a reference missing?

Generative AI is mentioned in the abstract as one of the technologies that makes data privacy even more important to pursue, why so? This is never explained after

The approach proposes to use semantic technologies to describe PETs and then use AI to select the best PET according to this description, what type of AI does this statement refer too?

How do distributed ledgers can help people regained ownership of their data?

Why would DataPrInTs make PETs context aware? Aware of legal context? Of the context of the user? This is not clear

What does the statement: “However, an important thing to further investigate and discuss concerning this approach is the sustainability of the visualisations themselves” what does sustainability refer too?  A reference might be helpful here.

Minor issue:
liscences is mispelled in the introduction

---

### Author Response · Authors · 2024-08-12
**Revision comments by the authors**

Dear reviewers and workshop organisers,

Thank you for your time reviewing our paper and your insightful comments. We have addressed them in the revised version of our paper.

1st Reviewer's comments:

The challenges in section 1 have been elaborated on and we have added references.

Regarding restructuring the paper, we have decided to keep our current structure as it first presents the reader with an overview of the proposed approach and then with more specifics on each building block. Hopefully, the added clarifications fix the issues raised.

We have double-checked the mention of Generative AI in the paper. It was mentioned in both the abstract and in section 2.2.5 of the paper as a tool that can be explored for generating personalised visualisations. We have now also added examples of this type of tool.

Regarding the question "How do DLTs help in regaining ownership of data?". The perspective of the DLT community (as also endorsed by the Solid community) is that individuals and communities place their data in their preferred location, be that their own machine or a trusted third party. DLTs allow one to immutably and publicly record encrypted metadata. As each DLT record is signed proof of data ownership and data integrity can easily be provided.

"Why would DataPrInTs make PETs context aware? legally? or user context?" The idea behind our proposed approach is that PETs will be combined with semantics (e.g. applying federated learning over multiple knowledge graphs; differential privacy over semantically enriched data). Data sharing in the use cases will be modelled with semantics, which are known to transform data into information and knowledge. The combination of PETs and semantics can therefore lead to more transparent AI which can help build more trust on both end-user and AI engineer sides.

"In the statement “However, an important thing to further investigate and discuss concerning this approach is the sustainability...” what does sustainability refer too?" Each individual has different comprehenison needs. Personalised visualisations can be a key to increasing the comprehension of decentralised data sharing for each user. However, having hundreds of visualisations for the same use case will affect the performance and scalability of the developed tool(s). Common visualisation elements that are positively viewed by all users can be filtered and reused. As a start, however, we need to know what visualisations help user's comprehend decentralised data sharing and how to best implement them.

2nd Reviewer's comments:

Missing definitions were added.

"too much weight is placed on grey literature..." - 10 of the references are from either the European Commission (e.g. on the new EU AI Act) or are from the UK's Information Commissioner’s Office. In these cases, we believe that it is imperative that we link our research to key national and international policies as they emerge.

Medicine has been removed as suggested.

A definition and reference for decentralised systems has been added.

"However, the process of implementing data and privacy governance in a decentralised setting is yet to be defined." The sentence has been extended.

Performance has been added as a challenge.

A definition for trust has been added.

"To ensure findable, accessible, interoperable, reusable (FAIR) [9] data principles are implemented, SemanticWeb technology, ... can be utilised" SW/RDF doesn't guarantee FAIR. The sentence has been edited.

"Following the proposed approach, a proof-of-concept prototype based on privacy-by design ..." Needs a prioritisation. Yes, we agree with the reviewer. As we define a more concrete implementation plan we will consider the aspects of our approach that need prioritisation. Depending on the research set up, front-end and back-end activities can be developed in parallel.

“’The tool’s evaluation in terms of its usability, ability to ease individual’s comprehension of decentralisation and increase trust in data spaces will help gather valuable insights on individuals’ perspectives of decentralisation and privacy." How will you measure trust?" This will be carried out via both qualitative and quantitative approaches. Quantifying trust in AI is currently a topic gaining popularity and metrics for it are being researched.

We have also looked at the paper the that the reviewer shared (https://www.sciencedirect.com/science/article/pii/S0160791X20303067). The paper argues that blockchains are 'confidence machines' rather than a 'trustless technology'. The main DLT literature advocates that the combination of decentralisation (users are free to be involved), consensus mechanisms for selection, transparency (every user gets a complete copy of the ledger), and cryptographic techniques supports trust between parties that do not trust each other. The most obvious example of this is the continuing use of Bitcoin which today has a market cap of $1.13 Trillion.

---

### Comment · Program_Chairs · 2024-08-16

Dear authors,
Thank you for addressing the comments of the reviewers.
We have a final improvement request towards accepting the paper for publication and presentation at the workshop.
Currently, the building blocks you present in Figure 1 do not match the blocks you put forward in Section 2.2.
Can you clarify better in the text how they are connected? As it is, the discrepancies can confuse readers.
Can you make these changes until next Tuesday, 20 August 2024 (5pm CET)?
Best,
NXDG 2024 organisers

---

### Author Response · Authors · 2024-08-19
**Updated manuscript revision**

Dear reviewers and workshop organisers,

We have now updated the titles of the building blocks in Figure 1 to mirror the subsection titles from section 2.2. The caption of the figure has been updated as well. We hope this helps avoid any confusion.

Kind regards,
The authors

---

### Decision · Program_Chairs · 2024-08-02

Conditionally Accepted